# Analysis of the Justice Component of a JEDI (Justice, Equity, Diversity, and Inclusion) Inventory in a College of Pharmacy

**DOI:** 10.3390/pharmacy12040118

**Published:** 2024-07-29

**Authors:** Chad W. Schulz, Jackson J. Dubas, Allison M. Dering-Anderson, Karen L. Hoff, Adam L. Roskam, Noah A. Kasbohm, Brady W. Holtmeier, Hannah L. Hansen, Kaitlyn L. Stukenholtz, Ashley N. Carron, Lindsey M. Tjards

**Affiliations:** Medical Center, College of Pharmacy, University of Nebraska, Omaha, NE 986145, USAkaitlyn.stukenholtz@unmc.edu (K.L.S.);

**Keywords:** college of pharmacy, diversity, equity, inclusion, inventory, justice

## Abstract

At the University of Nebraska Medical Center College of Pharmacy, a longitudinal project is underway to assess how the college is functioning in terms of keeping Justice, Equity, Diversity, and Inclusion (JEDI) at the forefront of the institution. This study is intended to showcase areas of excellence within the college and as a quality improvement exercise to show the institution potential areas in need of improvement. This process was also initiated because such assessments may soon become a requirement for colleges of pharmacy to earn full accreditation. Upon analyzing the Justice component of JEDI and the 32 justice-related ideas that were recommended for exploration and discussion, and further sub-categorized under the terms representation, curriculum and education, policies and procedures, support and resources, and college climate, useful data were discovered. Overall, the information found on representation, policy and procedure, and college climate was difficult to quantify as much of the information was subjective; however, this does not automatically discount this information from being useful. Information relating to curriculum and education was more quantifiable but may be underrepresented. Analyzing information found relating to resources was made possible by identifying readily available support offered at the college for faculty, staff, and students. In identifying these resources, the college was able to take note of any missing support that needed to be implemented to ensure justice was being maintained. This longitudinal process not only allows the college to see areas where they thrive, but it also highlights any shortcomings of the college while providing the institution with information to spark innovative ideas to strengthen and further promote justice.

## 1. Introduction

In the near future, a mandatory inventory of JEDI may become a requirement for colleges or schools of pharmacy to earn full accreditation [1,2]. While varying concepts of JEDI are being put into practice in organizations and businesses, the exact interpretations of JEDI and ways to quantify them remain gray areas [3,4,5,6,7,8,9]. Several different groups from the University of Nebraska Medical Center (UNMC) College of Pharmacy (COP) are working on this longitudinal inventory process. They include: the UNMC–COP Phi Lambda Sigma chapter, the UNMC-COP standing Curriculum and JEDI Committees, and fourth year pharmacy students who are completing their public policy Advanced Pharmacy Practice Experiential (APPE) elective rotation. In this report, references to UNMC are used to indicate the entire campus of the Medical Center. The University of Nebraska Medical Center (UNMC) campus is made up of six colleges (the College of Pharmacy, the College of Public Health, the College of Nursing, the College of Medicine, the College of Allied Health, and the College of Dentistry), two institutes (the Eppley Institute for Research in Cancer & Allied Diseases, the Monroe-Meyer Institute), and the graduate studies program. Through nearly fifty academic programs, UNMC educates more than 4400 students annually [10]. References to UNMC-COP indicate only the College of Pharmacy on the UNMC campus. The desired outcomes of this longitudinal project, in addition to meeting accreditation requirements, are to ensure that all members of the UNMC-COP (administration, faculty, students, support staff, etc.) feel seen, heard, and appreciated. This will allow all people involved to have the opportunities to cultivate growth individually as well as help advance the structures, processes, and resources of the college. We, likewise, hope to identify areas of weakness to allow for improvement at UNMC-COP.

The focus in this project report is the Justice component of the JEDI inventory. Justice is defined by the authors as: “the obligation to be fair in the distribution of benefits and risk” [11]. Multiple definitions of justice were reviewed with the authors ultimately choosing to define the term “justice” based on the definition found in Legal and Ethical Issues for Health Professionals 4th Edition [7,12,13,14].

In the initial report of this process, “Discussion of an approach to starting a JEDI inventory in a College of Pharmacy”, Schuff Zimmerman et al. attempted to create a strategic process to ultimately determine which components of an institution will be most vital in coordinating a JEDI inventory [11]. In that report, the authors defined the JEDI terms and created a listing of 148 ideal components of a JEDI inventory. This was done through literature reviews and brainstorming discussions to gather a comprehensive index of ideas highlighting what should be counted in an ideal JEDI inventory. This index contains 32 different components specifically relating to the Justice portion of JEDI.

Table 1 shows the 32 concepts included in the Schuff Zimmerman et.al. definition of an ideal JEDI inventory of Justice concepts. It describes which components were counted and how that count was accomplished. This project was not simulated or theoretical; this project was the first attempt to complete an inventory using the recommendations from Schuff Zimmerman et al. [11].

## 2. Project Description

### 2.1. Setting

This Justice component of the overall JEDI inventory was conducted in the UNMC-COP where students prepare to become experts in medication and pharmacotherapy. UNMC campus-wide data and policies were utilized when UNMC-COP specific data were not available or when UNMC-COP policies specifically deferred to campus policy.

### 2.2. Determining What to Inventory

The initial decision to pursue data collection for the JEDI project came from the UNMC-COP Curriculum Committee and UNMC-COP Beta Xi Chapter of Phi Lambda Sigma. An extensive literature review was undertaken to define each term included in the JEDI inventory and to craft a list of ideal inventory components. Focusing on Justice, the 32 “ideal” inventory components were assessed for the practicality of gathering the information needed. (see Table 1) [11,14,15,16,17,18,19].

### 2.3. Conducting the Inventory

Items were inventoried by analyzing course syllabi and college or campus-wide policies and procedures. Data were also gathered from existing mandatory student surveys and other resources made available on UNMC’s website [10]. Experts and leadership on the UNMC campus were asked about specific components of the inventory when there was difficulty locating the data.

## 3. Findings (The Inventory of Justice)

### 3.1. Representation (Inventory Items 1–3)

The first task related to representation was to quantify how many people from diverse groups are in decision-making roles. By identifying the leaders and faculty sponsors of student organizations, it was found that 87 (61.7%) of the 141 identified leadership positions were held by females; however, faculty sponsorship was predominantly male (77%). A breakdown of these roles can be seen in Table 2. Data on other diversity issues, race, ethnicity, religion, etc. are not readily available and are not assessed.

The next component assessed was the fair distribution of scholarships as designated by a committee in the college. Addressing these items was challenging, with item 3 deemed a conflict of interest for students by the By-Laws of the UNMC-COP. There is concern that a student who is eligible for a scholarship or award is unable to impartially serve on this committee. It is important to this project report that the items used for the inventory came from a publication intended to list ideal inventory components [11]. It is inevitable, given the premise of the initial work, that some items will not be countable.

There is a Scholarship and Awards Committee in place that identifies and distributes awards to students. Most scholarships have predetermined criteria that students must meet to receive said award. There are six members from faculty and staff from the UNMC-COP, and a representative from the UNMC community that make up the committee. Currently, the Scholarships and Awards Committee has five members, two men, three women, with one position unfilled. The College is made up of two departments (Pharmaceutical Sciences and Pharmacy Practice & Science). There are two members from Pharmacy Practice & Science, one member from Pharmaceutical Sciences, one member of student support from the College, and one member from financial aid from the UNMC campus. None of the members of the committee are BIPOC (black, indigenous, other people of color) at present [20]. All faculty representatives have more than 10 years of service on the faculty and the student services member has five years of service to UNMC-COP. Information about culture, ethnicity, and religion is not available for any member of the committee.

### 3.2. Curriculum and Education (Inventory Items 4–10)

All course syllabi from both required and elective courses from the 2022–2023 academic year were reviewed for many of the Justice components that fell under Curriculum and Education. There were 27 courses offered in the fall and 24 courses offered in the spring for a total of 51 course syllabi to assess. Figure 1 depicts how many courses included the following in their syllabi: a direct mention of a grade-challenging system (33.3%), exam review accommodations (17.6%), overall course point breakdown (92.2%), remediation policy (15.7%), course schedule with deadlines (62.7%), formal process for student feedback (0%), and at least one module that brought forth a discussion on justice (39.2%). Not all courses allow for grade challenges, and many may describe challenge policies and procedures separately from the syllabus.

UNMC-COP uses Canvas for course organization, and several course coordinators opt to post the syllabus, a grade or score challenge policy, a remediation policy, and a course calendar separately. Due to the difficulty in assessing what was discussed for every module in these courses, it was decided that this topic would be best quantified by counting the entire course to grasp if any topics of justice are being represented within the College of Pharmacy curriculum. It should be noted that, although zero course syllabi described a method for collecting and implementing student feedback, at the end of each semester students must complete course surveys.

Justice-related training offered by UNMC was also examined. First, we assessed whether justice awareness training was readily available for students, faculty, administrators, and staff for the UNMC-COP. Currently, there is only one readily available justice training provided to students. This is the mandatory Title IX student training which must be completed every 2 years as a Canvas module. Completion of Title IX training is not optional, students must complete this program. Administrative staff from the UNMC-COP report the current Title IX student module implemented on campus has 185 of 188 COP students (98.4%) up to date with the training. Ultimately, 100% of students must complete Title IX training or their records will be placed on hold until completion. Currently, faculty, staff, and administrators have mandatory Title IX training and mandatory Bystander training, enforced by Human Resources. The Bystander training for students is set to be released in academic year 2024–2025 and will also be mandatory. Each of these justice training opportunities is a UNMC campus-wide initiative and is not specific to the UNMC-COP.

### 3.3. Policies and Procedures (Inventory Items 11–23)

The inventory process began by analyzing the policies and procedures that have been established within the UNMC-COP. It was found that many of these policies and procedures were established for the entire campus of UNMC rather than being UNMC-COP-specific policies and procedures. Thus, these campus-wide procedures were counted for the remainder of this section.

First, policies and procedures that address and resolve conflicts of injustice were analyzed. As previously mentioned, the grievance resolution policy and procedure presented as a UNMC campus-wide procedure was the only one found. It states that each college is responsible for formulating its own conflict resolution procedure and publishing it. After conducting this research, with the help of other faculty members, it was possible to look at the standard operating procedures of each UNMC-COP standing committee, but we were unable to locate a grievance procedure specific to the UNMC-COP. An investigation of these campus-wide policies was then conducted to see how many formal policies have been established that directly address justice-related issues. There were 29 individual policies of 67 policies (43%) from UNMC’s online student policies and procedures list that directly mentioned justice and other synonyms or brought forth ideas about fairness [10,11].

Next, we were asked to quantify how many policies have been effectively communicated to students, faculty, and staff of the UNMC-COP. It was found that many policies have been formulated to optimize the experience that all people at UNMC have during their time on campus. It is difficult to determine if effective communication of these policies has been achieved. We then hoped to identify what percentage of the students, faculty, administration, and staff are aware of these policies. Much like attempting to quantify effective communication, quantifying awareness of these policies is difficult. These policies are publicly available and can be searched for and located online via UNMC’s website [10,11].

The policies on justice were explored to determine how many of these policies have been adequately assessed for effectiveness following their implementation. After reviewing the 29 policies that dealt with topics of justice, 6 policies had not been reviewed or updated since their date of implementation [10]. This equates to 20.6% of these justice-related policies not being reviewed for effectiveness. Figure 2 shows the breakdown of the 67 UNMC published policies: 38 policies do not deal with justice topics, 23 policies dealing with justice have been evaluated per campus mandate, and 6 justice policies have not been evaluated [10,21].

An analysis was completed to determine if there was an existing process in place for improving and evaluating existing justice procedures. Ultimately, it was found that a specific policy called “Process for the Review and Approval of Student Policies” explains how one would improve and evaluate said policies [21]. Part of this policy states that at a minimum, each policy is to be reviewed every 3 years. Leadership in student affairs and human resources were queried to determine if any students, faculty, administration, and staff had ever provided feedback on justice policies and procedures within the UNMC-COP. Members of the student senate and members of the policy approval process can provide feedback for the campus-wide policies and procedures. In addition to this, there are monthly town hall meetings that all students can attend. The location of the offices of student advocates, student affairs, and human resources and other contact information are publicly available.

There was a follow-up question that was then explored to see what the specific findings were for the feedback on the justice-related policies. It was found that, while feedback may have been given, there is no record of such feedback. According to the voluntary student satisfaction survey from the end of the 2022–2023 school year, 94% of respondents felt that students from all backgrounds were somewhat or very respected at UNMC. Within the UNMC-COP, 97% of respondents reported that they feel somewhat or very respected [10,22]. It should be noted that only 1007 students from the total UNMC student population and 68 students across all classes from the College of Pharmacy responded to this question in the survey. Data from the 2023–2024 survey showed 94% of respondents from the campus continued to feel respected, while specific data from the College of Pharmacy were not included in the 2023–2024 report [22,23].

An attempt was made to quantify how many instances of injustices against students, faculty, administration, and staff were reported. These data were not available. Personal interviews and personal experiences were evaluated to determine the methods used to make all people within the UNMC-COP aware of how and to whom to report injustices. Students have access to members of the administration or any faculty member they are comfortable talking with. These administrators and faculty can guide students and colleagues to the proper reporting mechanism. If the student is not comfortable with this approach to handling perceived injustices, then they can search the student policies and procedures page for the specific injustice they believe had been experienced and locate the proper person to report it to.

The last area explored was the potential impact of administrative decisions on diverse groups. There is a standard approval procedure for the implementation of new policies which includes a review from a multitude of different administrators and committees, but there is not a specific policy that considers the impact that institutional decisions have on diverse groups [21].

### 3.4. Support and Available Resources (Inventory Items 24–26)

Supportive resources on campus directed toward the maintenance of justice were counted to determine how many are available to staff and students. Examples of these individual resources include: the Assistant Vice Chancellor of Inclusion for the UNMC campus, the Director of JEDI within the UNMC-COP, and the Associate Dean of Student Affairs. It should be noted that during the time this manuscript was in process, searches for a new Assistant/Associate Dean of Student Affairs and a Director of JEDI had begun. Committees that are dedicated to supporting justice within the UNMC-COP were also assessed. Examples of committees that were identified in this search included the JEDI committee, the student discipline committee, and the grade appeals committee. The last area of support and resources available to students that was researched was the availability and utilization of legal support services on the UNMC campus during the previous three years. It was found that there are no legal services provided by UNMC for student utilization [24].

### 3.5. College Climate (Inventory Items 27–32)

The first issue we were asked to count was how many surveys are conducted to assess perceptions of justice at the college. The only surveys that are available to all students in the UNMC-COP are the end-of-the-year satisfaction surveys that have been previously discussed and individual course evaluations. There may be more surveys available to faculty and other students in specific roles, but the surveys described above are the only surveys found that are readily available to all students.

The next question asked about the number of incidents of discrimination reported. According to the Assistant Dean of Student Affairs, there have been five incidents of discrimination reported to him within the past 5 years. Another question investigated whether the college’s signage/symbols/art is inclusive of diverse groups. This is difficult to answer based on the complexity involved in determining what is considered diverse [20,25,26,27].

For temporary art/signage put up by faculty, administration, and public relations, many different concepts are uploaded onto visual displays to include diverse groups. These include holidays, religious practices, professional achievements, and organizational flyers. The final question concerning art/signage asked if the public relations promotional material of the college is reflective and supportive of diverse groups. Again, this is unable to be easily answered due to it being a subjective measurement of diversity.

The final questions posed in our initial work ask how many unconscious/implicit biases or incidents of microaggression were reported, but it was not possible to answer them due to lack of data. Offices within the UNMC-COP and the UNMC Division of Student Success, as well as the UNMC Assistant Vice Chancellor of Inclusion, were all unable to locate an answer or able to provide someone else to contact to further this search. With this many people unable to answer these questions, the assumption was that this number was zero.

## 4. Discussion

### 4.1. Representation

The findings from the representation component of the inventory were difficult to quantify overall due to the subjectivity of the definition of a “diverse group”. While data on diversity issues such as race, ethnicity, religion, etc. were not readily available and were not assessed, we observed that there was strong female representation in class officer positions and student-led organizations within the UNMC-COP. Interestingly all leaders included in this inventory identified as female or male. This will not always be the case, but for the period explored, it was. Agustin Fuentes argues that even within the identification of female or male that diversity exists [28]. We are not able to evaluate those sub-groupings currently. Further exploration of other diversity issues is planned for future JEDI inventories. To date, an IRB approval has been granted for an optional survey of some of these issues, but the work is not yet completed. While using gender to assess diversity is insufficient to be fully illustrative, it is a starting point for future work.

Lauchaud has explored the challenges in attempting to teach about diverse populations without inducing other biases [29]. We would agree and extend those thoughts to attempting to inventory diversity. It is our desire to contribute to this process to capture meaningful and actionable data. We believe that focusing on satisfaction and feelings of community will be more valuable than focusing on differences and varieties amongst the UNMC-COP community. Further discussion and exploration in this area are planned.

We investigated how the college ensures fair distribution of scholarships. There is a Scholarship and Awards Committee that identifies and distributes awards to students. It comprises six members of the faculty and staff from the College of Pharmacy. While the authors of the Schuff Zimmerman et al. article created an “ideal” inventory list, the authors of this manuscript had to consider the practicality of the specific inventory items when attempting to implement them in the real world [11]. Diversity within the Scholarship and Awards committee was ultimately determined to be irrelevant due to the blinded nature of the selection process and existing scholarship and awards criteria. Regardless of the make-up of the committee, the criteria drive the scholarship and awards presented. This decision was based on three main points: 1. Students are prohibited from serving on the Scholarship and Awards Committee by the UNMC-COP bylaws. 2. There are existing criteria which must be met to receive a scholarship or award. 3. Some decisions on financial aid and scholarships are not made within the UNMC-COP but are made by the UNMC or university-wide finance offices. The University of Nebraska System represents the University of Nebraska Medical Center, the University of Nebraska at Kearney, the University of Nebraska at Omaha, and the University of Nebraska at Lincoln. Some scholarships are awarded on a university-wide basis. Whether the scholarships and awards criteria aid in the fairness of award distribution can be argued. Most scholarship descriptions have financial need as a criterion, but none define financial need. Some scholarships and awards have specific criteria based on high schools attended, military experience, success in patient counseling, etc.

It can be argued that the blinded nature of the process and the criteria force the Scholarships and Awards Committee to be fair and impartial in granting these scholarships and awards. It can also be argued that failure to have criteria recognizing many of the components of diversity stops the process from being fully just or fair. Institutional restrictions on awards and scholarship criteria, prohibiting nepotism, or imposing structural bias, etc. also assist the Scholarships and Awards Committee in their pursuit of justice in awards. Knowing that the evaluation is blind, focusing only on criteria, gives confidence that bias is significantly reduced, if not eliminated.

### 4.2. Curriculum and Education

Based on what was found in relation to curriculum and education, it needs to be noted that the data may be underrepresented to what actual material, information, accommodations, and remediation possibilities students can utilize. For example, many professors and other course coordinators supplement a course schedule/deadlines or subjective grading rubrics in separate documents from their course syllabi. The data presented here correlate only to the information that was included in the syllabi. Since over half of the course syllabi do not include an explicit grade-challenging system (66.7%) and well over half of the course syllabi do not mention exam review accommodations (82.4%), these areas may be targets of improvement for the College of Pharmacy.

Remediation procedures were not discussed in 84.3% of course syllabi. While Figure 1 shows that only 62.7% of course syllabi included a class schedule and deadlines for the course, it must be considered that several professors hand out a separate schedule/deadline documents aside from the syllabus. The figure also shows that 100% of the course syllabi does not describe a method for collecting and implementing student feedback, but once again, we must consider the fact that there are required survey evaluations that must be completed by students to give feedback on professors and courses. On the other hand, some components of course syllabi were appropriately mentioned more often across the syllabi. Further evaluation of syllabi shows that while 92.2% of course syllabi included a point breakdown of the course, 60.8% of course syllabi do not mention any justice-related modules.

There is a course review form that is sent out to course coordinators and reviewed by members of a curriculum committee at UNMC-COP. This form includes a checklist for the course syllabus. This review form is not required to be completed every year for every course. It is distributed every three years to course coordinators and evaluated by the Curriculum Committee as defined in the UNMC-COP bylaws. The authors crafted a series of recommendations for the Curriculum Committee that may potentially improve the reporting of diversity, grading rubrics, and scoring challenges in course syllabi. The findings of this inventory process led the authors to recommend specific questions concerning those items we counted or attempted to inventory. It was reasoned that the course coordinator is the person in the best position to report on many of these issues, and incorporating that reporting into a standard course review would allow these sections of the justice inventory to be complete by 2027. A re-review of course syllabi is planned following decisions made by the Curriculum Committee with the recommendation of the JEDI Committee.

Our potential solution for addressing these shortcomings would be for the UNMC-COP to re-construct the syllabus rubric in the course review form to capture justice and diversity information directly from the course coordinators. While the UNMC-COP does excel in some components of their course syllabi, we must not be complacent. There is still room for improvement. This use of syllabi to implement and evaluate change as well as a defined route of communication is in keeping with the assessment plan published by Mercer University College of Pharmacy [30]. Interestingly, the best practices in syllabus design by Wagner, Smith, et al. comment on improved communication using well-crafted syllabi; there is no discussion concerning using the syllabi as evaluation tools in these kinds of inventories [31]. Christie, Coun, Bowen, et al. evaluated ethical reasoning in dental technician education [32]. They conducted their work without using syllabi. We believe that adding these core issues to the syllabus will lengthen it slightly, a concern for Wagner, Smith, et al. [31], but the benefit will far outweigh the added space. These updates will also allow for more data to be available for future justice inventories and quality improvement projects.

Although there are minimal justice-related training activities implemented campus-wide, there are plans to further develop training modules covering JEDI-related activities. It should also be noted that completing these training modules is and will continue to be required for students, faculty, and staff. When students do not complete these modules, holds are placed on their accounts which eventually inhibit them from being able to enroll in classes or graduate until modules are completed and holds are removed. If faculty and staff fail to complete these trainings, they may be referred to human resources. We have identified that the UNMC-COP has a solid foundation relating to justice training but continues to work to improve this area. Recent political focus on JEDI activities has the potential to complicate these efforts [27,33,34]. The authors have made recommendations to the UNMC-COP executive committee and the newly named Director, previously titled JEDI Director.

### 4.3. Policies and Procedures

Regarding policies and procedures, measuring effective communication is a subjective idea that is nearly impossible to quantify. Although policies and procedures are available for review by anyone, it is not possible to count “policy awareness”. The difficulty arises in determining when someone is considered aware of a policy. Is being aware of a policy indicative of knowing that the policy exists or does awareness correlate to the interpretation of the policy’s meaning and how it is ultimately applied to help UNMC affiliates? The best thing for UNMC to do is to continue to be transparent and open about their policies. UNMC’s policies are public and can easily be accessed online. Pharmacy students are told, during new student orientation, to report injustices to one of the Deans or any faculty member. One method to ensure students are aware of policies is to demonstrate specifically how to locate the policies and procedures online. Faculty and staff could be shown the policies and procedures webpage during their onboarding process. A reminder email could be sent out annually to current administration, faculty, staff, and students showing how to access UNMC policies.

Once again, it is difficult to quantify awareness of policies as this question has never been surveyed. Additionally, there may be groups in the UNMC-COP community who have a special need for assistance in understanding policies of this nature. Consider those with stress or other mental health challenges. Clark addresses some of these concerns in *“Imbalance”: Mental Health in Higher Education* [35]. Główczewski and Burdziej point out that previous experiences with fair treatment likewise have a significant impact on the application of policies meant to protect. If there is procedural fairness, or perceived procedural fairness, the student is more likely to be aware of and involved with college and campus processes [36]. The current UNMC campus-wide satisfaction survey is a possible tool to gather information on knowledge and understanding of policies. This UNMC survey, however, is voluntary with no plans to mandate completion currently.

While it was confirmed that UNMC has a grievance resolution policy in place, the UNMC-COP does not specifically have one for the college. Concern exists on the formal mechanism for communicating this process to every person at the UNMC-COP. The College of Medicine (COM) has their own student grievance policy that builds off the UNMC campus-wide grievance policy. The COM has a webpage that links to each of their student policies and procedures [37]. It may be beneficial for the COP to mirror the practice in the COM. This recommendation has been forwarded to the UNMC-COP executive committee for review. Our anticipation is that there will be the formation of an ad hoc committee to draft and promote a UNMC-COP policy on grievances. Included in our recommendation is including students and staff on this committee.

Several of the current UNMC-wide justice-related policies and procedures have never been updated nor reviewed. It was found, however, that there is a policy in place that states these procedures are supposed to be reviewed every 3 years. This shortcoming of the UNMC could be addressed by enforcing the “Process for the Review and Approval of Student Policies” policy. Another shortcoming within the policy and procedure area is there are currently no policies in place that require administrative decisions to report on the potential impact on diverse groups. This could be resolved by implementing a specific policy mandating this reporting. Further, enforcing the overall policy review every third year will allow for input from the UNMC community. The UNMC-COP members of the UNMC faculty senate have been approached to forward these recommendations.

With only 23% of UNMC students (36% of UNMC-COP students specifically) completing the student satisfaction survey, results may be skewed and not adequately represent how students feel while attending UNMC-COP. While the results in the student satisfaction survey are very positive, it may be inaccurate to assume there is no need for improvement. A solution to the low response rate is to send this survey out at the end of the school year as a required survey that must be completed by all students. This mandatory completion is not currently under consideration on the UNMC campus.

Review of specific survey questions to assure that students are queried regarding understanding policies, understanding reporting processes, and other data identified as not available or unable to be counted can improve the survey without adding an addition survey to the students’ survey burden [38]. It can also be postulated that highly dissatisfied students would be more likely to respond than those who are satisfied. This may imply that the current results are accurate.

### 4.4. Support and Resources

UNMC offers several supportive resources and committees on campus that are available to students, staff, faculty, etc. The intent of these resources and committees is to make certain that justice is being sustained. The assistant vice-chancellor of inclusion, JEDI committee, Director of the JEDI committee, Dean of Student Affairs, student discipline committee, and grade appeals committee are all agencies that can assist anybody within the UNMC-COP to ensure justice is supported throughout the college. There are currently resources and support for justice and other JEDI activities within the college and on the UNMC campus. The college must work proactively to continue to obtain these resources and guarantee their accessibility. Legal support services, however, are not available to UNMC-COP administration, faculty, staff, and students. The University of Nebraska system does have a College of Law that resides on the Lincoln, Nebraska campus. There is a student legal services (SLS) program offered at the University of Nebraska Lincoln (UNL) that offers free representation or legal advice to currently registered UNL students [14]. It would be interesting for the COP to explore any potential options of working with the College of Law to be able to offer these legal services to their students.

The challenges of addressing microaggressions, implicit bias, etc. and the corresponding lack of any data for evaluation has led the Division of Student Success to explore if there are no reports because no one tracks this or because students do not know how to report concerns appropriately. Changes in tracking and additional documentation are expected in the future.

### 4.5. College Climate

There are no specific surveys for students that focus only on justice. The UNMC-COP requires end-of-school-year surveys and course evaluations to allow students to give feedback and express concerns. Some of these end-of-school-year surveys do include concepts of justice or other JEDI concepts. It may be beneficial for the COP to offer a specific survey focusing solely on JEDI concepts. This recommendation has been forwarded to the new director of the formerly titled JEDI department at UNMC-COP.

Each of the UNMC-COP standing committees are dedicated to maintaining a fair, safe, and academically prosperous environment for the success of all UNMC-COP students. While the previous Dean of Student Affairs was able to cite five reported incidents of discrimination that had been brought to his attention over the past 5 years, the UNMC-COP was unable to determine how many incidents of unconscious/implicit biases had been reported. Several different offices were contacted to find out the proper reporting procedures for these specific incidents. Some of the offices contacted included those within the College of Pharmacy and Division of Student Success, and the Assistant Vice Chancellor of Inclusion. All information was gathered via personal interview or email. There are no published data, or even a central repository for such data, causing some concern. It may be naive to assume that there are zero incidents of microaggression and unconscious/implicit bias that occur within the UNMC-COP. This leads us to further question if students are unaware of where or how to report an incident. A proposed solution would be for the UNMC-COP to decide on and appoint a dedicated person or committee within the college who will take on the task of being the sole entity that handles the reporting of microaggression and unconscious or implicit bias incidents so that there is an ability to track and a record of these issues, allowing the UNMC-COP administration to take preventative or corrective action.

The authors have determined that UNMC-COP marketing materials, art, and signage are neither just nor unjust. They may demonstrate a clear intent to be fully representative of diversity, but weaknesses may be discovered, but this is not a justice issue in our opinion.

### 4.6. Additional Post-Inventory Changes

The UNMC-COP has created a dedicated part-time faculty position to support the concepts of justice, equity, diversity, and inclusion. This position will be essential in supporting ongoing efforts to achieve the UNMC-COP goals in these areas.

The authors have requested that the UNMC-COP Executive Committee formally recognize and celebrate the three outstanding accomplishments identified.

The collaborative nature of this longitudinal process makes student input and involvement a key by focusing on APPE students interested in policy and Phi Lambda Sigma, the student leadership fraternity. It also merges the JEDI and Curriculum standing committees into this endeavor.The careful adherence to criteria in the Scholarship and Awards Committee removes as much subjectivity and bias as possible. This standard operating procedure for the Scholarship and Awards Committee should be replicated. Where possible and appropriate, blinding the identity of the student/faculty/staff and basing decisions on pre-determined and publicized criteria is a laudable goal. Interestingly, this process of blinded application and criteria-based evaluation is the process chosen for issuing invitations to join the UNMC-COP Beta Xi Chapter of Phi Lambda Sigma.Recognition of the results of the UNMC student satisfaction survey is also appropriate given the positive results regarding well-being. This recognition has the potential to increase future response rates.

### 4.7. Identified Areas for Improvement at UNMC-COP

Improving information without increasing survey burden.More clearly defining reporting channels for concerns and making recommendations for improvement. Education in reporting options and processes will be necessary.Creating a data repository and responsibility for maintaining the information to allow for all future inventories and quality improvement work to have access to needed data.

The authors recognize several weaknesses in this inventory report. Firstly, justice does not exist in a vacuum. Each of the concepts of justice, equity, diversity, and inclusion are interdependent upon one another. All future work on this project will focus on the totality of JEDI, not singular concepts. We chose to evaluate diversity based solely on gender. This does not clearly represent the diverse people at UNMC-COP.

## 5. Conclusions

This manuscript examines the findings of the initial inventory of the justice component of the JEDI inventory at the UNMC-COP. Upon analyzing the 32 ideal components that fell under the justice portion of the JEDI inventory, it must be noted that quantifying many of these components was not fully attainable. This is largely since many of them were subjective in nature. Although quantifying some of this data was unachievable, the UNMC-COP still has a great deal of useful information that can be analyzed and applied towards making positive reinforcements to better their institution and gain an understanding of what revisions need to be made in the future to fully live up to and foster their believed definition of justice. The information from this manuscript can also serve as a point of reference to hold ourselves accountable and learn from our past while striving to be better in the future. We plan to use this information to improve the UNMC-COP and UNMC as a whole and to implement changes for the success of all students.

The process of creating a justice, equity, diversity, and inclusion inventory can push institutions beyond simply fostering the idea of an inventory for the sake of counting, and into a position of working for improvement. While it may become a requirement for colleges or schools of pharmacy accreditation soon, this process is not simply about acquiring the stamp of approval and moving on. It goes far beyond that and is truly about creating a conducive learning and work environment for all students, faculty, and staff. It will allow universities to become more accepting and authentic institutions that offer safe, nurturing, and inclusive spaces to learn.

## Figures and Tables

**Figure 1 pharmacy-12-00118-f001:**
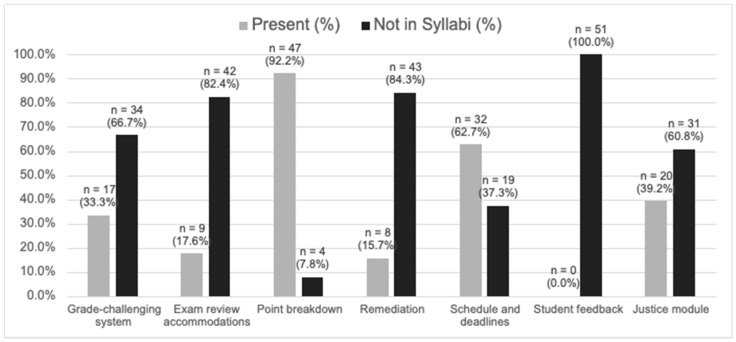
Curriculum and education components in course syllabi.

**Figure 2 pharmacy-12-00118-f002:**
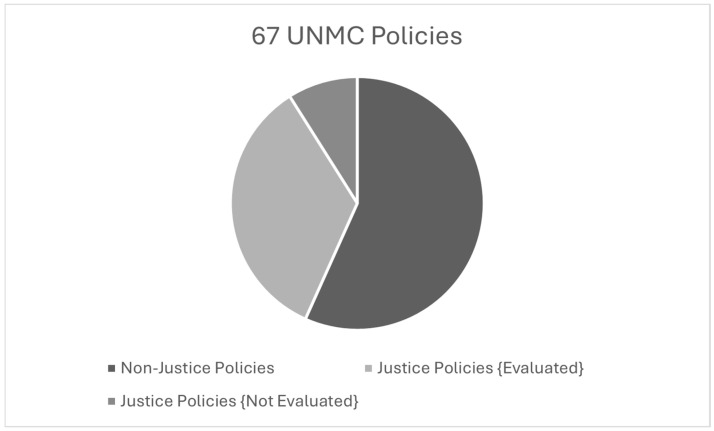
The 67 UNMC policies reviewed.

**Table 1 pharmacy-12-00118-t001:** Potential items of Justice for a JEDI Inventory at a College of Pharmacy, adapted from Schuff Zimmerman et al. [11].

Justice Inventory Items	Counted	Description
**Representation**		**3.1**
How many people from diverse groups are in decision-making roles	Yes	Leadership by Gender see Table 2
2. Are there designated organizations in the college designed to ensure the fair distribution of scholarships	Yes	Scholarships and Awards Standing Committee
3. If so, what percentage of leadership within this organization is made up of diverse staff and students	Yes	Students are prohibited from serving on the committee. Committee membership evaluated for diversity.
**Curriculum and Education**		**3.2**
4. How many courses or modules address topics of justice	Yes	See Figure 1
5. What % of college syllabi include an explicit grade-challenging system A process to challenge grades from laboratory or simulation assignments A process to review and challenge exam/quiz questions A process for providing feedback for all grade/exam question challenges	Yes	See Figure 1
6. What % of college syllabi include a transparent grading system Objective point breakdown Rubrics for subjective scoring Deadlines Remediation	Yes	See Figure 1
7. What % of college courses allow for a student review of the system in 6	Yes	See Figure 1
8. What accommodations are provided to students during exam reviews and/or question challenges 1:1 reviews memory aides for students who are unable to view a copy of their results during challenges	Yes	No accommodations have been requested for exam reviews/question challenges, to date
9. Is Justice awareness training readily available for: Students Faculty Administration Staff	Yes	Title IX Training is readily available Bystander Training is readily available
10. What % of people have taken mandatory justice awareness training Students Faculty Administration Staff	Yes	See Section 3.3
**Policies and Procedures**		**3.3**
11. How many formal policies deal directly with justice issues	Yes	29 out of 67 of UNMC’s campus-wide policies. See Figure 2
12. How many policies have been communicated effectively to students, faculty, staff, administration	No	Difficult to quantify effective communication
13. Has the college developed a procedure to address conflicts and injustice	No	The UNMC-COP has not developed a specific college policy. The UNMC-COP has adopted the campus-wide grievance resolution procedure
14. What policies are in place to measure/take into consideration the potential impact of administrative decisions on diverse groups	No	Difficult to quantify or determine awareness of these policies
15. Have policies on justice been assessed for effectiveness	Yes	23 out of the 29 UNMC campus-wide justice-relating policies have been reviewed, while 6 policies have not been reviewed since their implementation
16. Is there an existing process for improving and evaluating existing justice procedures	Yes	Standard approval procedure in place for implementation of new policies, but no policy that specifically considers impact that administrative decisions have on diverse groups
17. What % of students, faculty, staff, administration are aware of these policies	No	Difficult to determine and quantify awareness
18. Have students, faculty, administration, and staff provided feedback on justice policies and procedures	No	No specific questions in satisfaction surveys were identified. No comments or concerns have been documented.
19. What were the findings of this feedback if any	No	There is no identified feedback mechanism
20. How are individuals in the college made aware of who/where to report said injustices	Yes	Students urged to speak to Deans/faculty members or can look at the student policies and procedure page on UNMC’s website to find the correct person to report to
21. How many examples of injustice can students, faculty, administration, or staff cite	No	Data were not quantified at the UNMC level
22. What % of students completed the justice survey	Yes	1007 students from the total UNMC student population (~23%) and 68 students from the COP (~36%)
23. What % of the students report that their experiences and perspectives were respected/valued within the college	Yes	97%
**Support and Resources**		**3.4**
24. How many resources are dedicated to ensuring justice	Yes	Assistant Vice Chancellor of Inclusion for UNMC, Director of JEDI within the UNMC-COP, Dean of Student Affairs
25. How many committees are dedicated to supporting justice	Yes	JEDI Committee, student discipline committee, grade appeals committee
26. How many students have used legal support services in the previous 3 years	No	Lack of legal services—unable to be quantified
**College Climate**		**3.5**
27. How many surveys are conducted to assess perceptions of justice at the college	Yes	End-of-year surveys and individual course evaluations
28. How many incidents of discrimination have been reported	Yes	5 incidents within the last 5 years
29. How many incidents of unconscious/implicit bias have been reported	No	Unable to identify reporting procedures, nor to locate any records of reports
30. How many incidents of microaggressions have been reported	No	Unable to identify reporting procedures, nor to locate records of reports
31. Is the college’s signage/symbols/art inclusive of diverse groups	No	Deferred to the Diversity component of this project
32. Are the public relations, promotional materials for the college reflective and supportive of diverse groups	No	Deferred to the Diversity component of this project

**Table 2 pharmacy-12-00118-t002:** Gender distribution within leadership roles.

Leadership Role	N = 141	Male, *n* (%)	Female, *n* (%)	Total, *n*
Faculty Advisor		10 (77)	3 (23)	13
Class President		4 (100)	0 (0)	4
Class Vice-President		3 (75)	1 (25)	4
Class Secretary		0 (0)	4 (100)	4
Class Treasurer		0 (0)	4 (100)	4
Class Social Chair		3 (33)	6 (67)	9
Class Historian		1 (12)	7 (88)	8
Class Fundraising Chair		1 (8)	11 (92)	12
Class Tech Chair		4 (50)	4 (50)	8
Student Senate Representative		2 (100)	0 (0)	2
Student Organization President		4 (33)	8 (67)	12
Student Organization President-Elect		5 (62)	3 (38)	8
Student Organization Vice President		2 (50)	2 (50)	4
Student Organization Secretary		2 (20)	8 (80)	10
Student Organization Treasurer		4 (40)	6 (60)	10
Miscellaneous Student Organization Position *		9 (31)	20 (69)	29

* Includes positions that are student organization-specific and are not replicated in other student organizations.

## Data Availability

The raw data supporting the conclusions of this article will be made available by the authors on request.

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
