# Peer review of "Analysis of the Justice Component of a JEDI (Justice, Equity, Diversity, and Inclusion) Inventory in a College of Pharmacy"

_pharmacy, 2024, doi:10.3390/pharmacy12040118_

Round 1
Reviewer 1 Report
Comments and Suggestions for Authors
Thank you for the opportunity to review the manuscript titled " Analysis of the Justice Component of a JEDI Inventory in a College of Pharmacy " .
The authors described the component of Justice, Equity, Diversity, and Inclusion (JEDI) inventory at the UNMC College of Pharmacy. They based on the analysis of the 32 ideal components that fell under the justice portion of the JEDI inventory.
The manuscript is interesting because it shows the areas of excellence within the college and some which need to improve. There are not many good articles about the problem.
There are some issues that the authors should address before the manuscript can be considered for publication.
The following are my comments describing these issues.
1.The title of the manuscript
The authors used the abbreviation JEDI in the title. This abbreviation is used for various names for example JEDi: Java Essential Dynamics Inspector. Please consider whether it would be better to use the full name in the title and not just the abbreviation.
2. Abstract
Line 17 and Line 30- The text from the introduction was rewritten into the abstract along with references (1,2,3) that should not be included there.
3. Introduction
The authors use abbreviations very often in the manuscript. However, these abbreviations are not always explained for example
Line 38- Several different groups from the UNMC College of Pharmacy (COP) are…
Or
Line 350- UNMC offers several supportive resources…. Do you mean the same The UNMC College of Pharmancy
Please correct the abbreviations throughout the manuscript and explain them. And wherever possible, please provide the entire names, for example at the beginning of the sentence.
4. Results
Please consider adding a table or diagram that will present the results of your research more clearly than the text itself.
5. Discussion
The discussion requires significant improvement.
In the discussion the authors did not refer to the results of the work of other authors.
(Please do not describe the results again in the discussion, but if possible, compare them with the results of other authors).
6. References
Please add references.
In this article, authors should provide creative solutions to problems, especially those aspects where they failed to get results. The manuscript becomes more interesting.
Reviewer 2 Report
Comments and Suggestions for Authors
This project outlines a longitudinal study at the University of Nebraska Medical Center's College of Pharmacy, aimed at evaluating the institution's commitment to Justice, Equity, Diversity, and Inclusion (JEDI). This study serves as a model for similar academic institutions to review and enhance their JEDI policies. The authors are encouraged to consider the following suggestions to further improve the quality of this manuscript.
Perform statistical analysis on the data presented in Table 2 and Figure 1 to determine if there are statistically significant differences.
Create a bar chart or a table to represent the findings from Section 3.3, "Policies and Procedures" (Inventory Items 11-23).
Provide more concrete and actionable recommendations for implementing justice components in the academic setting. For example, to increase the student satisfaction survey response rate, consider incorporating the survey into mandatory coursework or offering incentives for completion.
Develop sample questions related to Justice component to be included in the annual course feedback surveys.
Authors have addressed the lack of previous data on various aspects of JEDI. However, authors can suggest a methodology for archiving data related to incidents and reports.
Reviewer 3 Report
Comments and Suggestions for Authors
Thank you for the opportunity to review your manuscript regarding the evaluation of your JEDI assessment tool. While I applaud the focus for the work, much of this reads like a commentary about an attempt to utilize your developed inventory without a specific research aim or systematic process of reporting information on the requests from the inventory. Unfortunately, much of the information was difficult to follow and it was very unclear how results were collected, the process for analysis, and ways for the reader to evaluate the reliability and validity of the findings. I greatly appreciate and acknowledge the attempt by the authors; however, there are significant concerns about the presentation of information.
One of the important limitations of this work is that much of it was focused exclusively on gender or visible elements of identity, which is very surface level for JEDI work. Unfortunately, the evaluation using the inventory did not look at various identities (race, cultural, ethnic, religious, political, romantic, etc.) or their intersectionalities to clarify how this tool can support programs. Instead, it more so illustrated the challenges with this work and that may be the highlighted point.
I have provided comments below for opportunities to update the manuscript, as desired.
INTRODUCTION
- UNMC is not spelled out the first time the acronym is used, please include it (it's listed out in 2.1 instead, but would be better upfront)
- Stylistically, I prefer shorter paragraphs for ease of reading. A small suggestion is to start "the focus in this project report" as it's own paragraph. It'll draw more attention to the emphasis of this work and the definition of justice.
- It would be helpful to clarify the purpose of the project was to complete a mock implementation of the inventory (as far as I am understanding)--if that is an incorrect interpretation, then greater clarity would be very useful in this section.
- The paragraph that starts "In "discussion of an approach", I am not sure the article title needs to be provided, instead it could be simplified to something about previous research with the necessary citation. I also think this paragraph should come before the paragraph about the focus for the study to improve the flow.
- I am confused about the "description" and "counted" columns in Table 1 and how to interpret them. In the leading paragraph it's stated "what items should be counted in an ideal inventory", is that meant to refer to what's included in the inventory or is it referring to an tabulation or frequency of items? Are these two columns necessary? I like just having the list of questions and it would make the table more manageable from a cognitive load perspective for the reader. You could include an asterisk on items that are "not counted" to indicate that status (which I am interpreting as they were not in the final inventory). Alternatively, it may be ideal to have the full list in an appendix and the non-counted items referenced in this manuscript not be listed in the table to prevent confusion.
METHODS
- There is limited to no information about how policies and other items were identified as being related to "justice' for evaluation by your team, especially as it relates to policies, course syllabi, etc. The definition provided in the introduction does not provide enough clarity about how justice may be referenced in these different environments and examples would be valuable in the introduction for broader audiences that may not be familiar with this work.
SECTION 2.3
- This is likely carry over from the previous section, the statement "items that were counted were inventoried"
FINDINGS
- The section regarding the scholarship and awards committee is rather confusing and difficult to understand--for example, what was the conflict of interest for students referenced in the first statement?
- Section 3.3 refers to a lot of data collection methods on how items were quantified or evaluated, but that is not described in the methods section to evaluate the reliability or validity of the data generated. For example, the attempts to quantify the awareness of policies and procedures and the resolution was that they could be searched for.
- Much of the results from other sections reads more as a report about the current situations within the program rather than interpretation of how a person may use the inventory to evaluate the criteria, reporting, etc. Much of this highlights the nuances and difficulties of using these types of inventories, which unfortunately counters the argument that these tools are necessary and useful.
DISCUSSION
- Overall, the section again reads more as a report on the program rather than a discussion about how the tool is related and applicable to work for other programs. The authors consistently cite the issues with using the inventory, which suggests it was not properly designed for practicality or utility by the user.
Reviewer 4 Report
Comments and Suggestions for Authors
I applaud the students to create the idea supported by faculty to assess one’s own college for current justice college and university status followed by thoughtful interpretation of the data to determine needs, assessments and best practices going forward. Having all parties included in this type of research, analysis, and future change is critical. It was not clear if any staff member also participated. Much data were collected to create a foundation for reflection and advancement. Having an inventory helped divide the concepts into categories and focus the presentation, interpretation and discussion of the findings. Such a reflection could create the impetus for other colleges to do so as well.
Overall the manuscript is written well but many items in the findings section were more related to discussion, with some of these concepts repeated again in discussion. Thus for the revision, move all discussion and future ideas to the discussion section. At times the paper sounded more like just looking at justice for students as some topics were not related/evaluated for faculty and staff, so wonder if a few more details for faculty and staff are required as the goal was justice for the college. As the college only measures gender, a limitation is justice by race, ethnicity and other diverse groups was not included. I did not see any discussion to determine justice by other demographics that are associated with injustice in the future, which would be critically important. Collecting this data could improve your inner reflection and growth going forward.
Major
Ln 43 – do you plan on including alumni and preceptors?
Ln 45 – should the assessment be a major and separate category also?
Ln 49 - might be good to include the definition you used and why. Not sure the need of reference 3 justice definition if you did not use it.
Ln 60 and 90 – why did you limit diversity to just gender? As this is a forward data collecting project, why did you not ask for race, ethnicity, and religion of the various leaders with an option to prefer not to say if you were really looking for justice issues? Please provide the percent of your total students per year, faculty and staff for gender and other demographics collected. Some of these data are collected at the university and college level. At my college, we have group data for the diversity of our classes, faculty and staff, which I think you might also have.
Ln 92 Table 2 does well at representing gender for student organizations, what about faculty and staff leadership positions?
Ln 108 – the new term is BIPOC. Minority is no longer a preferred term as it connotes lower status.
Ln 156 – you reviewed policies and procedures for student justice, what about faculty and staff?
Ln 166 – how are these policies incorporated into orientations, and yearly reports, college/department agendas? I would think this could be quantified. So the processes but not the effectiveness part of this inventory item could be addressed.
Ln 170 – a bit of a stretch to say a review of a policy is equal to review of its effectiveness if data related to the policies are not collected.
Ln 223 – do you have a record of the number of students bringing their own legal counsel to justice disputes?
Ln 233 – do you have any data if these incidents were resolved using justice principles with just outcomes?
Some discussion occurs in the various findings sections and could be moved there or eliminated as it is already in the discussion section. I found much duplication between findings and discussion so please determine where the interpretation of the data will be placed and minimize duplication.
Ln 256 – Yes strong female representation in supportive class officer positions but no female presidents and only one female vice president. This does not qualify as potentially just representation if female students are predominantly in supportive leadership positions. Seems this would be an area of exploration to see if female students running and not winning or feeling low self-esteem/imposter syndrome to not run for the top offices. To fully understand these concepts the percent of females in each class should be given. I suspect like most colleges of pharmacy that female:male ratio is around 70:30. And in this case, there should be more female presidents and vice presidents. In the discussion I was expecting that future research/evaluation/assessment would look at BIPOC leadership positions in faculty, staff, and students, however it does not appear this is planned but critical to the issues of justice and explicit and implicit biases.
Ln 265 and 272 – diversity would not be irrelevant. The data could/should have fostered a task force/committee charge to look at just availability of scholarships to students from diverse backgrounds and identify any gaps to pursue funding opportunities for new scholarships.
Discussion – could add capturing and analyzing justice by more demographics/social determinants.
Minor
Justice is not a proper noun so not sure it needs to be capitalized.
Ln 21 – data is a plural noun, so needs a plural verb. Datum is singular.; some changes also needed throughout text and tables
Ln 38 – need to spell out UNMC
Ln 41 – need to spell out APPE
Ln 60 3.2.10 – the description says see 3.2, which item 10 is already a part of. Seems a better description of percents of groups taking mandatory training. Of note, mandatory training is not described
Ln 60 3.3.11 – as the paper describes a college of pharmacy, how many unique college of pharmacy policies and procedures address justice outside of university policies and procedures?
Ln 60 3.3.12 – do you have any of this in new faculty and staff member orientations, new student orientations?
Ln 60 3.3.14 – seems this is asking about follow-up after decisions. Are any records kept, who assesses compliance, end of year reports, any oversight?
Ln 60 3.3.17 – seems a simple survey to ask about familiarity within various groups with a wide range of policies and procedures would be doable. AACP survey asks some of these questions but their questions might not be specific to your inventory, but is a means to access awareness.
Ln 60 3.4.24 – you list personnel resources, what about financial resources, consultants?
Ln 60 3.4.26 – I would think the Dean would know if any student brought in outside legal counsel to help with justice issues.
Ln 60 3.5.27 – what question relates to justice on your individual course evaluations
Ln 60 3.5.28 – did any of these incidents involve legal counsel?
Ln 60 3.5.31 and 32 – understand this is being evaluated by the diversity committee, can you list their findings? The originators of the JEDI inventory felt signage and promotion were also a justice issue.
Ln 63 – are any of the other medical center programs doing this also? Is this a university initiative or just the college of pharmacy?
Ln 59 – is this adapted or word for word the same? Should the adaptation statement be a footnote vs. a title component?
Ln 82 – please define the existing surveys used.
Ln 116 – 199 – list information in table or text, but not both
Ln 128 – students must complete surveys or are they given the opportunity?
Ln 161 – not clear what optimize experience means.
Ln 295 – does the form include all the items you were evaluating?
Ln 311 – and now time to tackle the hard issue do faculty, staff and students practice the JEDI concepts and behaviors in the required training. Being knowledgeable does not fully equate to action. Great first step but how do you plan to go further?
Ln 314 – effectiveness can be hard to ascertain but distribution of information can be assessed.
Ln 348 – I think you mention above that the survey is required, but I could be confusing your various surveys.
Ln 355 – personnel resources stated above but no other resources discussed. Do you have other resources that should be added? Do you have an anonymous reporting of JEDI problems in your college? What does your JEDI director or vice-chancellor of inclusion do?
Ln 358 – do you not have an ombudsman for assistance?
Ln 373 – do you have a process in place to evaluate outcomes and justice for the incidents reported?
Ln 436-463 - some references appear not to be in the correct format or have space, punctuation and or capitalization errors.
Text – review text to eliminate extra spaces between sentences.
Comments on the Quality of English Languagesome minor changes needed, "data" is a plural noun; justice is not capitalized, and extra spaces between some sentences
Round 2
Reviewer 1 Report
Comments and Suggestions for Authors
The authors improved their manuscript in accordance with the reviewer's comments.
Reviewer 3 Report
Comments and Suggestions for Authors
Thank you for providing the revisions to the submission--no additional comments from me.